# Probabilistic Tsunami Hazard Analysis of Inundated Buildings Following a Subaqueous Volcanic Explosion Based on the 1716 Tsunami Scenario in Taal Lake, Philippines

Kwanchai Pakoksung *, Anawat Suppasri and Fumihiko Imamura

International Research Institute of Disaster Science, Tohoku University, Miyagi Prefecture 980-0845, Japan; suppasri@irides.tohoku.ac.jp (A.S.); imamura@irides.tohoku.ac.jp (F.I.)
* Correspondence: pakoksung@irides.tohoku.ac.jp

**Abstract:** A probabilistic hazard analysis of a tsunami generated by a subaqueous volcanic explosion was performed for Taal Lake in the Philippines. The Taal volcano at Taal Lake is an active volcano on Luzon Island in the Philippines, and its eruption would potentially generate tsunamis in the lake. This study aimed to analyze a probabilistic tsunami hazard of inundated buildings for tsunami mitigation in future scenarios. To determine the probabilistic tsunami hazard, different explosion diameters were used to generate tsunamis of different magnitudes in the TUNAMI-N2 model. The initial water level in the tsunami model was estimated based on the explosion energy. The tsunami-induced inundation from the TUNAMI-N2 model was overlaid on the distribution of buildings. The tsunami hazard analysis of inundated buildings was performed by using the maximum inundation depth in each explosion case. These products were used to calculate the probability of the inundated building given the occurrence of a subaqueous explosion. The results from this study can be used for future tsunami mitigation if a tsunami is generated by a subaqueous volcanic explosion.

**Keywords:** subaqueous volcanic explosion; probabilistic tsunami hazard of building; tsunami simulation

## 1. Introduction

Tsunamis generated by subaqueous volcanic eruptions represent only a few of all recorded tsunamis, which are more commonly generated by earthquakes or landslides [1]. However, a tsunami can be generated by the eruption of a subaqueous volcano, and water surface displacement is triggered by its eruption, thereby generating tsunami hazards, which are principally interesting for evaluating and forecasting because there is a lack of geographical, observational, and contributory data. For example, with the subaqueous eruption of volcanoes in Taal Lake in the Philippines, as shown in Figure 1, large-scale resonance within the lake is generated due to tsunamis triggered by subaqueous volcanic eruptions. [1–3].

Taal volcano has produced 33 historical eruptions, and five notable eruptions generated tsunamis in the lake in 1716, 1749, 1754, 1911 and 1965 [3,4]. The 1716 eruption was a subaqueous volcanic explosion that was located in the southern basin of the caldera and near the southern point of Taal Island. The decompression of the gas trapped inside the magma lifted the water in the lake, forming a wave that impacted the shores. The maximum inundation in the southwestern area reached a height of approximately 17 m on land [5]. The 1749 eruption clearly produced a wave in response to the violent VEI4 (volcanic explosivity index) phreatomagmatic eruption [3,4]. The eruption vent was located north of Taal Island, and the wave impacted the northern shores of the lake. The 1754 event killed 20 people through building collapse of roofs induced by the weight of the volcanic ash deposited on the roofs [4]. Thus, observations clearly indicated the existence of a tsunami wave during this event. A phreatomagmatic to phreatoplinian eruption occurred

in 1911 on Taal Island. A tsunami with a wave height of approximately 3 m affected villages on the western shore of the lake, and 20–50 people (depending on sources) were drowned [3]. The 1965 eruption was also a phreatomagmatic eruption that generated a new crater on the southwestern flank of the island [4]. In total, 190 to 355 fatalities occurred when tsunami waves impacted the western shore of the lake, with a wave runup of 4.7 m above the mean level of the lake [6]. In 2020, the Taal volcano erupted on January 12, and the eruption occurred in the main crater, with a lava fountain height of 500 m [7]. Ash from this eruption spread around the volcano, reaching distances of approximately 10–15 km, but there were no reports of a tsunami in Taal Lake during this event. According to the eruption in Taal Lake in 2020, we have concentrated on future tsunami hazards in this area for future tsunami disaster management. To mitigate future tsunami disasters, the probabilities of various scenarios that could occur in the area have been estimated [3,8,9]. In this study, we aim to determine the probability that tsunami hazards will impact urbanizing areas in response to subaqueous volcanic explosions in Taal Lake for stakeholders and policymakers.

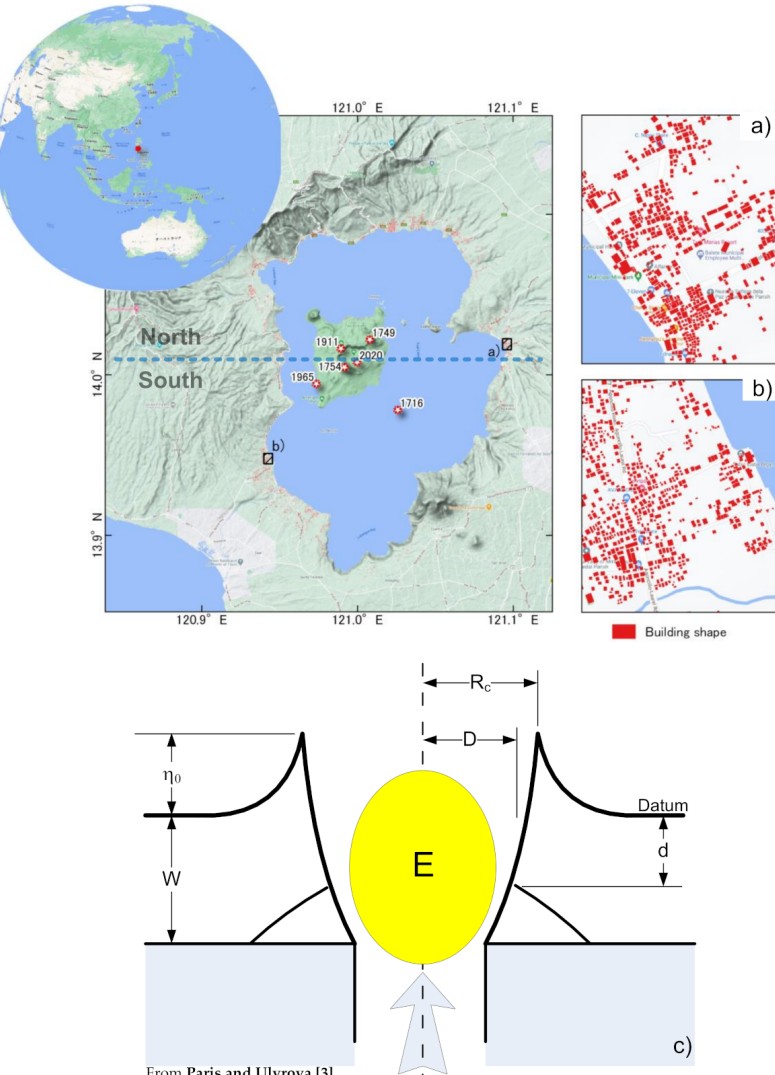

**Figure 1.** Location of Taal Lake and the history of eruptions in the lake; the blue dashed line is assumed to be the border between the north area and south area of the lake. (**a**) Example location of buildings in the eastern region and (**b**) example location of buildings in the western region. (**c**) Mechanism of initial water displacement from subaqueous volcanic explosion, $E$: explosion energy; $d$: explosion depth; $D$: diameter of vent area; $R_c$: diameter of water cavity; $W$: total water depth around the volcano; $\eta_0$: initial water surface displacement.

The characteristics of a tsunami generated by a subaqueous volcanic explosion are shown in Figure 1c. For example, in the case of the 1716 eruption in Taal Lake, tsunamis are controlled by several physical parameters, such as water depth, size of the eruption vent, depth and energy of the explosion, and magma-water interaction, which are used to define the explosion itself (as explained in detail in Le Mehaute [10]; Kokelaar [11]; Wohletz [12]; Mirchina and Pelinovsky [13]; Duffy [14]; Le Mehaute and Wang [15]; Kedrinskii [16]; Egorov [17]; Morrissey et al. [18]; Paris and Ulvrova [3]). The explosion forms an initial crater, resulting in a similar cavity at the water surface with a cylindrical bore. The cylindrical bore expands radially to form the leading wave, followed by a wave trough. The initial water surface displacement, corresponding to the maximum height of the bore, can be empirically calculated as a function of the explosion energy (see details in Le Mehaute [10]; Sato and Taniguchi [19]; Goto et al. [20]; Paris and Ulvrova [3]). The initial water displacement downward follows the upward displacement and forms a steep cone in the center of the bore. The cone collapses and generates a second bore, as reproduced by the experimental explosion and numerical models in previous studies [3,15,16,21,22].

Several methods of probabilistic tsunami hazard analysis (PTHA) have been developed during this decade [23]. PTHA is based on the probable values of tsunami parameters, such as tsunami height, flow velocity, flow depth, runup height and inundation extent, to be exceeded at a particular location within a given period. The PTHA can integrate geographical, historical and experimental data on both tsunamis and tsunami sources using statistical and numerical methods. However, PTHA methods for a volcanic source of tsunamis are more limited than those for an earthquake source [23]. Here, tsunamis are produced by a volcanic source, and this work focuses on a conditional PTHA in which the volcanic mechanism has the potential to generate a tsunami from a subaqueous explosion.

In this study, tsunamis generated by a subaqueous explosion were considered under different scenarios in Taal Lake. The scenarios vary in the size of the explosion, with the eruption location confined to that of the 1716 event, which represents one of the largest submarine explosions in the lake. Previous studies have focused only on tsunami hazard assessment and mapping [8,23], while tsunami hazard assessment involving building exposure has not been studied. Thus, to address this gap, we first prepared a probabilistic tsunami hazard analysis of building exposure based on different scenarios of subaqueous explosion-induced tsunamis affecting the coastal area around the lake.

## 2. Methodology

The methodology presented in this paper is based on three elements: tsunami source calculation, tsunami simulation and tsunami hazard estimation, as shown in Figure 2.

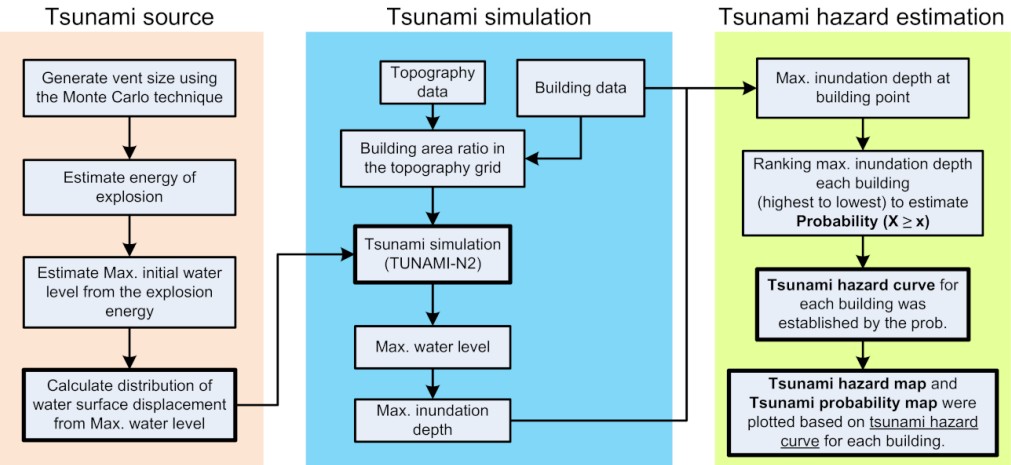

**Figure 2.** Flow chart of this study used to achieve probabilistic tsunami hazards.

### 2.1. Model of Subaqueous Explosion and Tsunami Generation

We selected 99 different sizes (vent diameter varied from 100 to 1000 m) of simulated explosions in Taal Lake with the eruption location corresponding to the location of the 1716 subaqueous eruption. The energy of an explosion can be estimated with the empirical formula provided by Sato and Taniguchi [19]:

$$E = 4.45 \times 10^6 \, D^3, \tag{1}$$

where $E$ is the explosion energy in Joules and $D$ is the diameter of the vent area in meters. The explosion energy generates an initial surface displacement, as described by Torsvik et al. [21] and Ulvrova et al. [22,24]. A parabolic cavity with a vertical steep rim, representing the size of the water cavity ($R_c$) in meters, can be calculated based on the explosion energy as follows:

$$R_c = 0.0361 \, E^{0.25}, \tag{2}$$

The estimation of the maximum initial water level is related to the explosion energy and water depth of the explosion $d$ in the empirical function presented by Le Mehaute and Wang [15]:

$$\eta_0 = c \, E^{0.24}, \tag{3}$$

where $\eta_0$ is the vertical initial surface displacement corresponding to the maximum initial water level in meters, $E$ is the energy of the explosion in Joules and $c$ is a constant that is a function of the energy of the explosion and water depth [15]. The constant is assumed to have one of two values: (1) $c = 0.0143$ if $6.15 \times 10^{-4} < d/E^{1/3} < 1.85 \times 10^{-2}$ and (2) $c = 0.0291$ if $E < 6.15 \times 10^{-4}$. The distribution of water surface displacement from the crater ($\eta$) is based on the maximum initial water level ($\eta_0$) using the following equation [15,25]:

$$\text{if } r_e \leq R_c, \ \eta = \eta_0 \left[ 2\frac{r_e}{R_c} - 1 \right] \tag{4}$$

$$\text{if } r_e > R_c, \ \eta = 0 \tag{5}$$

where $r_e$ is the distance from the explosion center in meters.

### 2.2. Numerical Simulation of a Tsunami

To obtain the tsunami inundations for different explosion sizes, a numerical tsunami simulation was driven with the TUNAMI-N2 model [26,27]. The TUNAMI-N2 model was first developed at Tohoku University to model tsunami propagation and inundation on land and operates using the nonlinear theory of the shallow water equation, which is solved using a leap-frog scheme. The nonlinear shallow water equation is presented as Equations (6)–(8), wherein the finite difference method is applied to run the nonlinear equation with a friction of surface represented by Manning's roughness coefficient:

$$\frac{\partial \eta}{\partial t} + \frac{\partial M}{\partial x} + \frac{\partial N}{\partial y} = 0, \tag{6}$$

$$\frac{\partial M}{\partial t} + \frac{\partial}{\partial x}\left(\frac{M^2}{D}\right) + \frac{\partial}{\partial y}\left(\frac{MN}{D}\right) + gD\frac{\partial \eta}{\partial x} + \frac{gn^2}{D^{\frac{7}{3}}}M\sqrt{M^2 + N^2} = 0, \tag{7}$$

$$\frac{\partial N}{\partial t} + \frac{\partial}{\partial x}\left(\frac{NM}{D}\right) + \frac{\partial}{\partial y}\left(\frac{N^2}{D}\right) + gD\frac{\partial \eta}{\partial y} + \frac{gn^2}{D^{\frac{7}{3}}}N\sqrt{M^2 + N^2} = 0, \tag{8}$$

where $\eta$ is the water level, $M$ and $N$ are the fluxes of water in the $x$ and $y$ directions, $D$ is the total depth, $g$ is gravitational acceleration and $n$ is Manning's roughness coefficient. The preparation process was performed for the bathymetry grid for the tsunami propagation and inundation simulations.

In the TUNAMI-N2 model of inundation in an urban area, the buildings have a high impact on the flow resistance in the estimation of the inland water level. The model uses

only topography data, so the results might have low accuracy because the terrain has several obstructions such as buildings in the real world [28]. The resistance impact can be applied by the composite equivalent roughness coefficient from surface types and buildings estimated by using Equation (9), which has been explained with the concept of the flow resistance induced by large ground objects [29,30]. This equation is related to the density of buildings and the flow depth, and the roughness increases as the flow depth increases:

$$n^2 = \frac{(100 - \theta)}{100} n_0^2 + (CD/2gW)\frac{\theta}{100} D^{4/3} \tag{9}$$

where $n_0$ is Manning's roughness coefficient for surface types without buildings, $\theta$ is the density of buildings in the computation grid as a percentage (the method used to estimate this parameter is presented in Appendix A), $CD$ is the drag coefficient and is set to 1.5 [31,32], $W$ is the horizontal scale of the buildings in meters and $D$ is the inundation depth at the building point for each time step in the simulation in meters.

All numerical simulations were performed by using the TUNAMI-N2 model based on a 1145 × 1171 grid with a resolution of 30 m. The grid was obtained by combining bathymetry and topography; the bathymetry was digitized from Ramos [33], and the topography was collected from the Shuttle Radar Topography Mission (SRTM, [34]). A computational domain was used to perform a constant-grid tsunami simulation, which yielded 11 million equations and unknowns to be solved at each time step of 0.01 s, and the Open Multi-Processing (OpenMP) platform was applied to achieve a more rapid computation time. At the boundaries, the open sea featured nonreflective boundary conditions and the shore areas had no specific boundary conditions for wet/dry fronts [26]. Note that the area without buildings was assigned a constant Manning's coefficient of 0.025.

Building shapes were collected from OpenStreetMap (OSM) though Quantum Geographic Information System (QGIS) open-source software, as shown in Figure 1a,b. The collected building data were limited to only shape and location, without the character of height. In this study, we analyzed buildings in an area close to the shore of Taal Lake within a distance of approximately two km. The shapes of the collected buildings were overlaid on the grid cell to estimate the building area percentage in each grid cell. The building percentage was used to estimate the roughness in Equation (9), and the building location was used to determine the maximum flow depth for probability hazard analysis in Section 2.3.

### 2.3. Probabilistic Hazard Analysis

For each scenario, the maximum inundation depth in all inland areas was estimated by subtracting the topography (from SRTM) data from the simulated maximum water level. The maximum inundation depth from the 99 scenarios was overlaid on the building locations from the OSM data to determine the inundated buildings and maximum inundation depth at the center of the building shape. The maximum inundation depth of each inundated building was used to create the hazard curve for each simulation case.

Tsunami intensity was measured by selecting the maximum inundation depth recorded at each inundated building. The hazard curve was established for 13,548 buildings that were ranked in decreasing order on the basis of the maximum inundation depth of each inundated building. A log-normal distribution was assumed to represent the hazard curve. The parameters of the distribution, the mean ($\mu$) and standard deviation ($\sigma$) of the logarithmic value, were estimated based on the simulation results. $\mu$ was estimated to be the average value of the log(maximum inundation depth), while the standard deviation of the log(maximum inundation depth) was set to be $\sigma$ for this study (following Pakoksung and Takagi [35]). Here, specific inundation depths ranging from 0 to 50 m were considered with a depth bin of 0.1 m, and the probability of exceedance was computed by estimating the parameter for each inundation depth in the given range to establish the hazard curve.

The probability value of the hazard curve for each inundated building was plotted to present the hazard in a spatial map. The mapping was performed by inputting the

probability value into the property of the building shape that was collected from OSM. This method for mapping the probability value to the building shape was performed by following Nofal and Van de Lindt [36].

## 3. Results

### 3.1. Tsunami Wave Generation and Inundation Extent

Numerical tsunami simulation results from 99 scenarios are presented in this section. The maximum tsunami amplitude and inundation area from numerical modeling were calculated from terrain data with a grid size of 30 m for a simulation time of 60 minutes. The 99 scenarios of explosions with vent diameters varying from 100 to 1000 m (which control the probability of occurrence) are presented in Figure 3a. Based on Equation (1), the vent sizes corresponded to explosion energies of $1 \times 10^{15}$ to $35 \times 10^{15}$ J, as explained in Figure 3a. Figure 3b presents the relationship between the explosion energy and the maximum initial water level based on Equation (3). The calculated initial water levels varied from 25 to 135 m for the lowest and highest explosion energies of the assumed scenarios, respectively.

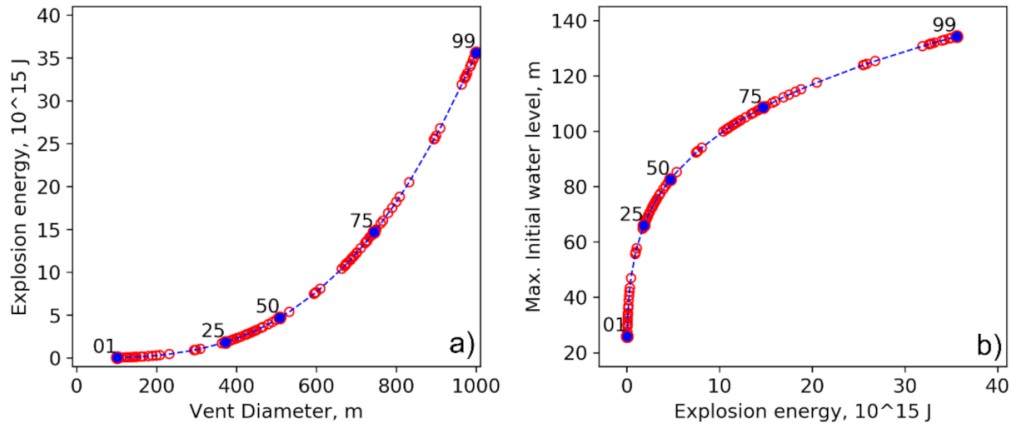

**Figure 3.** Scenarios of subaqueous volcanic explosions: (**a**) the relationship between vent size and explosion energy, (**b**) maximum initial water level related to explosion energy.

Tsunami generation was based on the maximum initial water level and size of the water cavity, and the diameter of the cavity was calculated by Equation (2). The distribution of the initial water level was identified as a parabolic shape that was generated with Equations (4) and (5). Figure 4 presents the tsunami generation in the 99th scenario with a vent diameter of 1000 m and a maximum initial water level of 135 m; the propagation is shown at different times, namely, 0, 10, 30 and 60 s. The tsunami initially features a parabolic shape, after which the second wave forms and moves from the center of the explosion point.

Figure 5 presents the maximum water level after 60 minutes in the tsunami model for three different scenarios, namely, the 1st, 25th, 50th, 75th and 90th scenarios, as well as the maximum amplitude distribution in the lake. The difference between the maximum water level and the topography reflects the inundation extent and flow depth based on the runup process. The middle and right columns in Figure 5 show the extent of inundation in the five different scenarios and its overlap with the buildings.

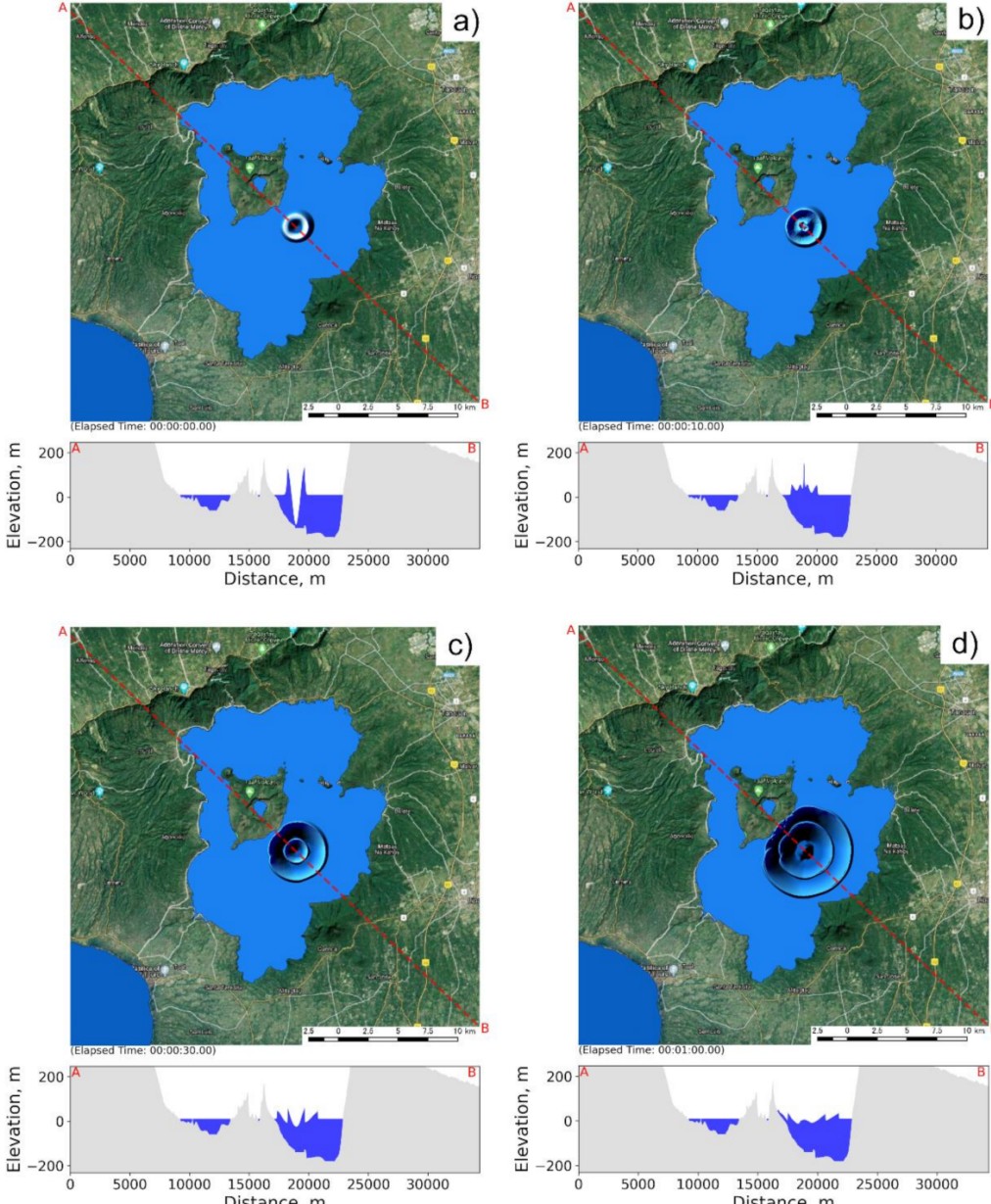

**Figure 4.** Tsunami generation: (**a**) 0 s, (**b**) 10 s, (**c**) 30 s, and (**d**) 60 s.

The inundation depth at each building point was determined for each building occupancy. Figure 6 shows the tsunami characteristics with the number of inundated buildings represented by the histogram for five examples of scenarios. There are 29,889 residential buildings in the coastal area of Taal Lake, and the analysis revealed that 13,548 buildings were impacted by the 99th scenario. In all scenarios, the highest number of inundated buildings corresponded to low inundation depths between 1 and 10 m. For example, the peak of the 1st scenario was at a water depth of approximately 1 m, while the peak of the 99th scenario was at a water depth of approximately 2 m.

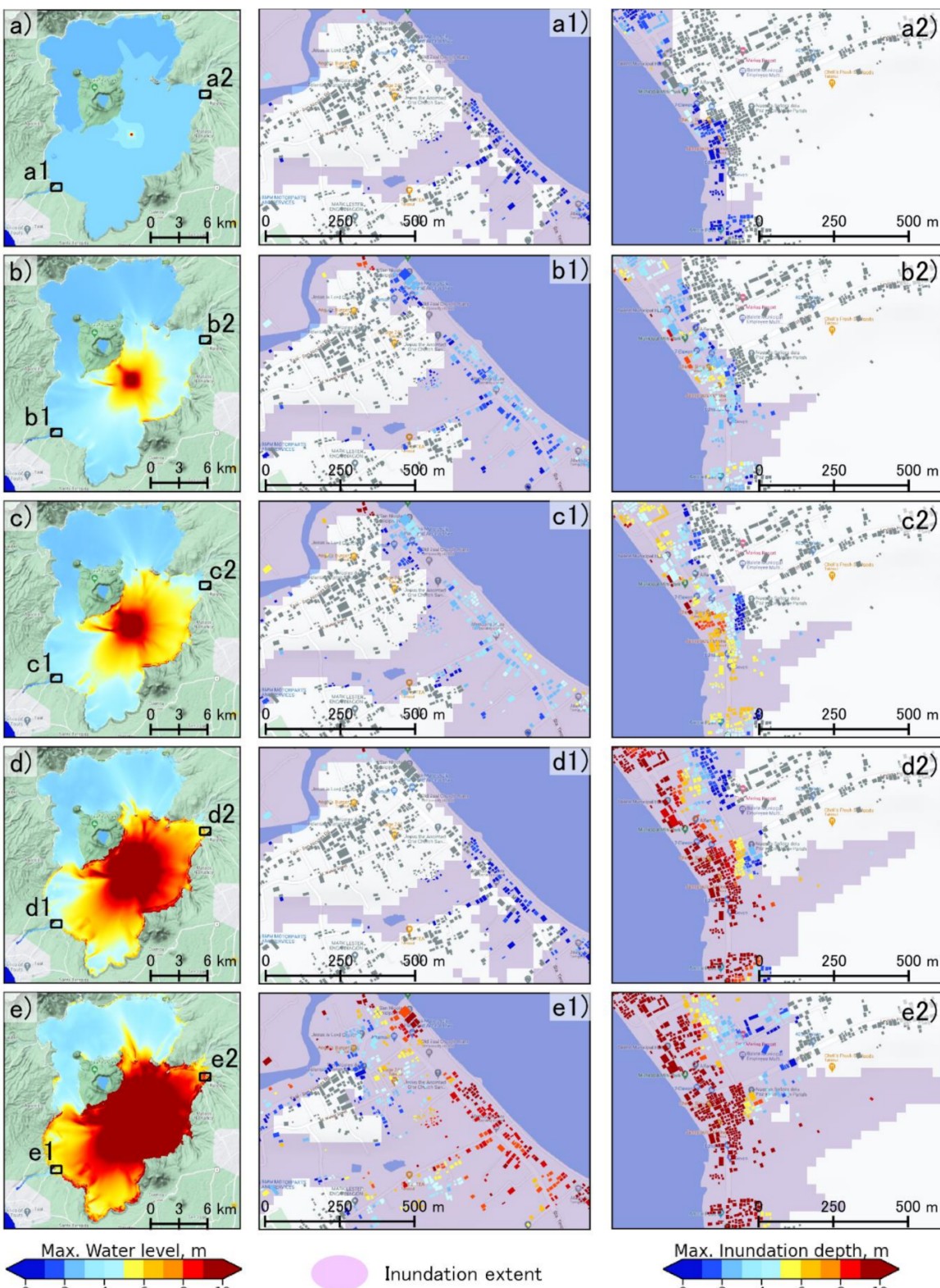

**Figure 5.** Maximum water level, flooded area, and an overlay on the building to present the inundation depth at building: (**a**) 1st scenario (explosion energy of $0.374 \times 10^{15}$ J and associated initial water level of 25.8 m), (**b**) 25th scenario (explosion energy of $1.84 \times 10^{15}$ J and associated initial water level of 65.9 m), (**c**) 50th scenario (explosion energy of $4.69 \times 10^{15}$ J and associated initial water level of 82.5 m), (**d**) 75th scenario (explosion energy of $14.7 \times 10^{15}$ J and associated initial water level of 108.5 m) and (**e**) 99th scenario (explosion energy of $35.6 \times 10^{15}$ J and associated initial water level of 134.2 m).

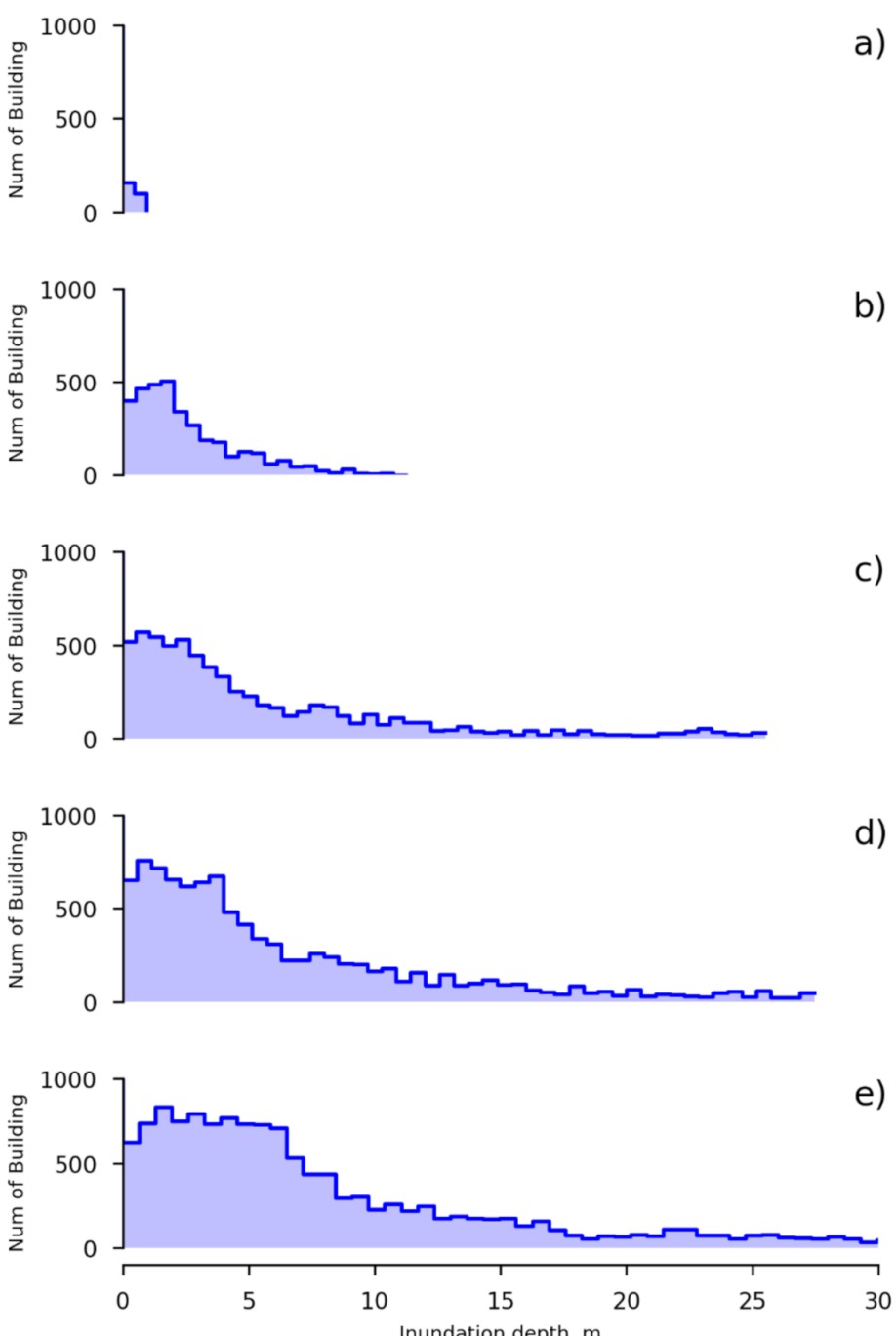

**Figure 6.** Number of buildings represented in each inundation depth (flow depth at the building point) range to present the hazard characteristics, with examples of (**a**) the 1st scenario, (**b**) 25th scenario, (**c**) 50th scenario, (**d**) 75th scenario, and (**e**) 99th scenario.

### 3.2. Probabilistic Condition Hazard Assessment of Inundated Buildings

An inundation map of buildings for some example scenarios is shown in Figure 5 in the middle and right columns. This reveals that the inundation depth of the buildings increased depending on the level of energy explosion magnitude. The number of inundated buildings in each scenario was used to develop the hazard assessment, which is the goal of this study.

There were 6494 and 7054 hazard curves in the north and south, respectively, and the north and south were divided by the middle of Taal Island within Taal Lake at a latitude of 14 degrees (see Figure 1). Figure 7 presents the tsunami hazard curve at each region.

The southern area presented a higher exceedance probability than did the northern area, based on the mean curve at the probability of 1/1000. The inundation depth of 1.0 m in the southern area corresponded to a probability of 1/2, where the exceedance probability of more than 5 m corresponded to a probability of 1/5. A limitation of this study is that the tsunami simulation was performed without consideration of the effect of a coastal structure. A coastal structure can be affected by a tsunami in a particular location on the corresponding hazard curve, but this study presented the hazard curves for only larger features, such as islands or regional mountains, to determine the regional impact presented by the mean hazard curve.

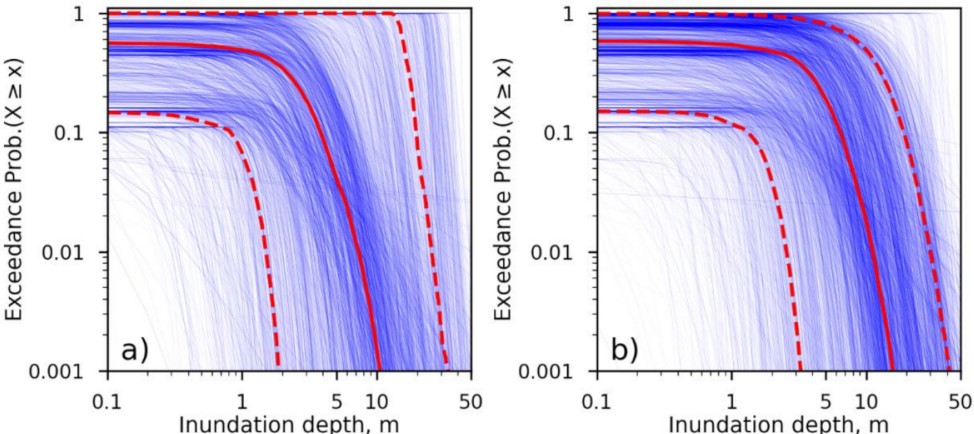

**Figure 7.** Individual tsunami hazard curves of the buildings at specific building points (light blue) including the mean and 10th–90th percentile confidence interval for an inundation depth of 0.1 m. (**a**) northern coastal area, and (**b**) southern coastal area.

The tsunami hazard map was derived from the tsunami hazard curve shown in Figure 7, as shown by Figures 8–10, for probabilities of 1/1000, 1/500 and 1/100, respectively, for the three example areas. The inundation depths for the presented probabilities are represented by different colors at the building locations, while the nonimpacted buildings are not shown with colors. The color map has ten different colors on a gradient from blue to red. The lowest hazard (<1 m) is shown in blue, whereas the highest hazard is shown in red (>25 m). For the whole study area, the hazard map for probabilities of 1/1000, 1/500 and 1/100 is summarized in Figure 11. Most of the tsunami-impacted buildings had a depth of 1 to 30 m for probability of 1/1000 and 1/500, while the 1/100 probability corresponded to tsunami-impacted buildings with depths ranging from 1 to 10 m in the north and south.

The hazard can alternatively be shown in terms of the probability of exceedance as a percentage ($p \times 100\%$). Figures 12–14 present the probabilities of experiencing a tsunami with an inundation depth of more than 0.5, 1.5 and 3.0 m, respectively, at all the inundated building points, in the three example areas. A depth of 0.5 m can be used to initiate tsunami damage, and a depth of 3 m can be identified as major tsunami damage [37–39]. The probabilities of exceedance for the selected inundation depths are presented by colors on a map of the buildings, while the nonimpacted buildings are not colored. The color map has ten different colors on a gradient from blue to red. The lowest probability of exceedance (1–10%) is shown in blue, whereas the highest probability of exceedance (90–99%) is shown in red. For the whole study area, the probability map for depths of 0.5, 1.5 and 3.0 m is summarized in Figure 15. Most of the tsunami-impacted buildings had a probability of 10 to 50% in the north, and 60 to 99% in the south, of experiencing tsunami inundation at these depths.

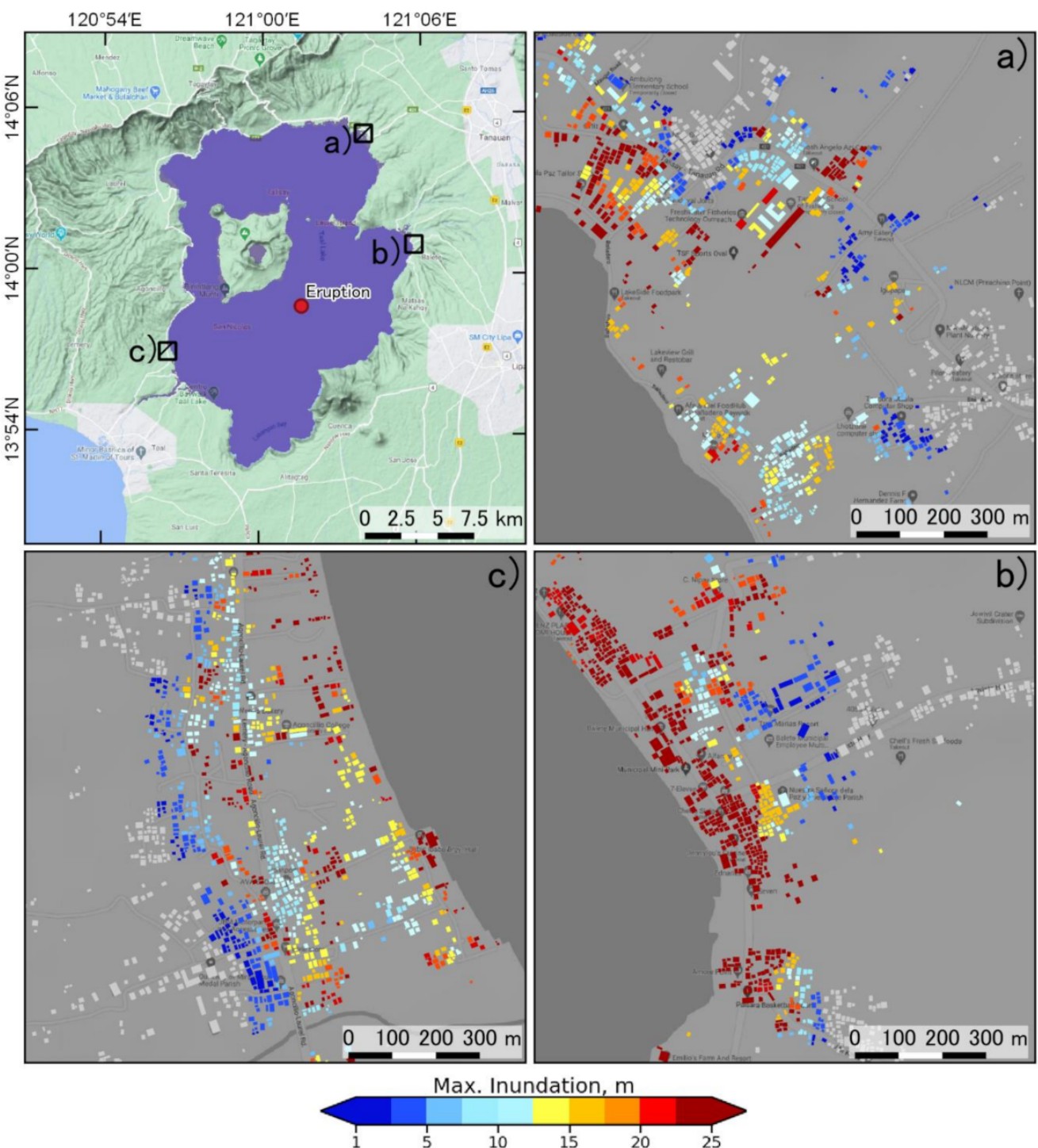

**Figure 8.** Tsunami hazard map showing the maximum inundation depths for the buildings with a probability of 1/1000 in some example areas. (**a**) northern coastal area, (**b**) coastal area near to the eruption point, and (**c**) southern coastal area.

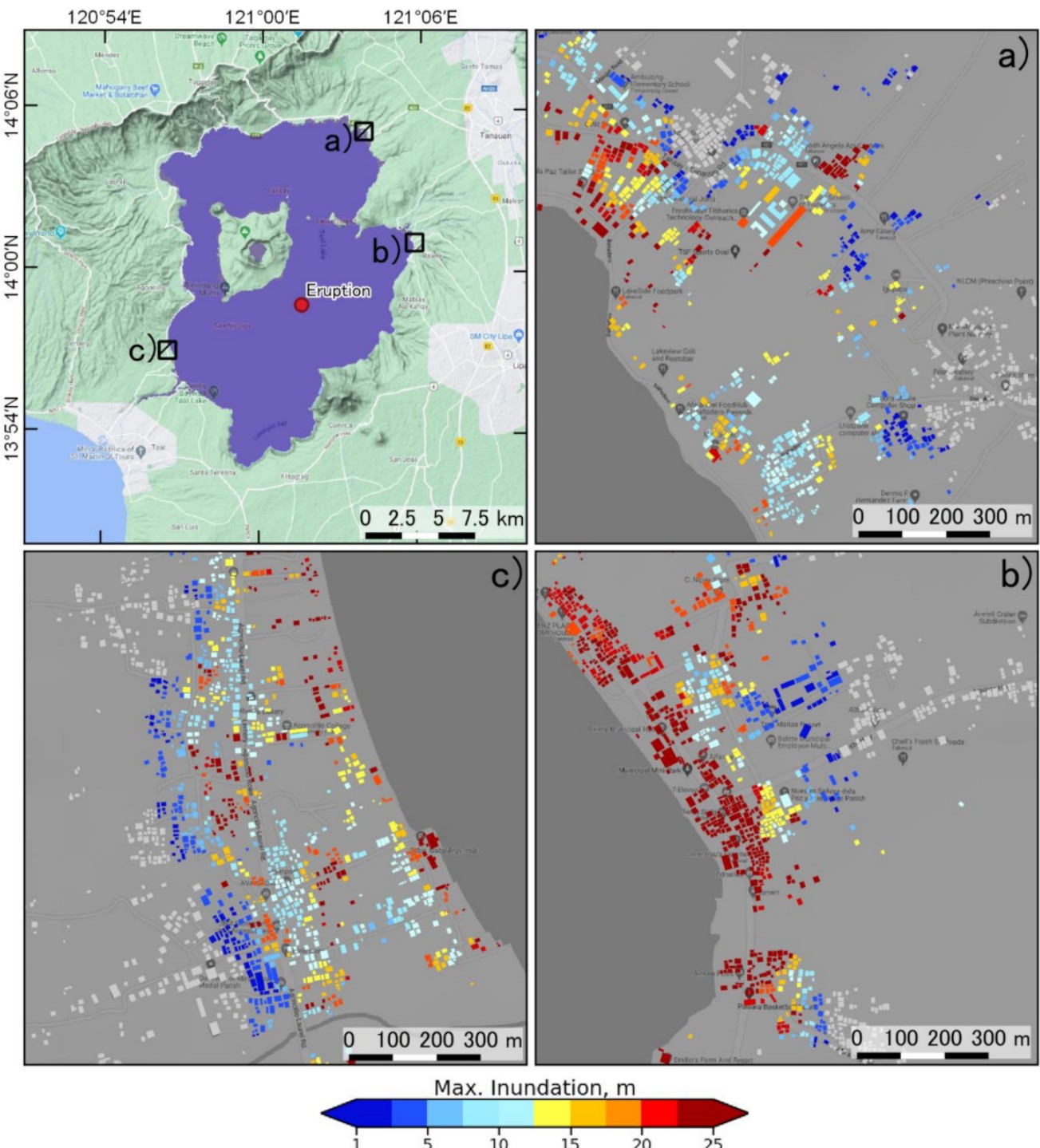

**Figure 9.** Tsunami hazard map showing the maximum inundation depths for the buildings with a probability of 1/500 in some example areas. (**a**) northern coastal area, (**b**) coastal area near to the eruption point, and (**c**) southern coastal area.

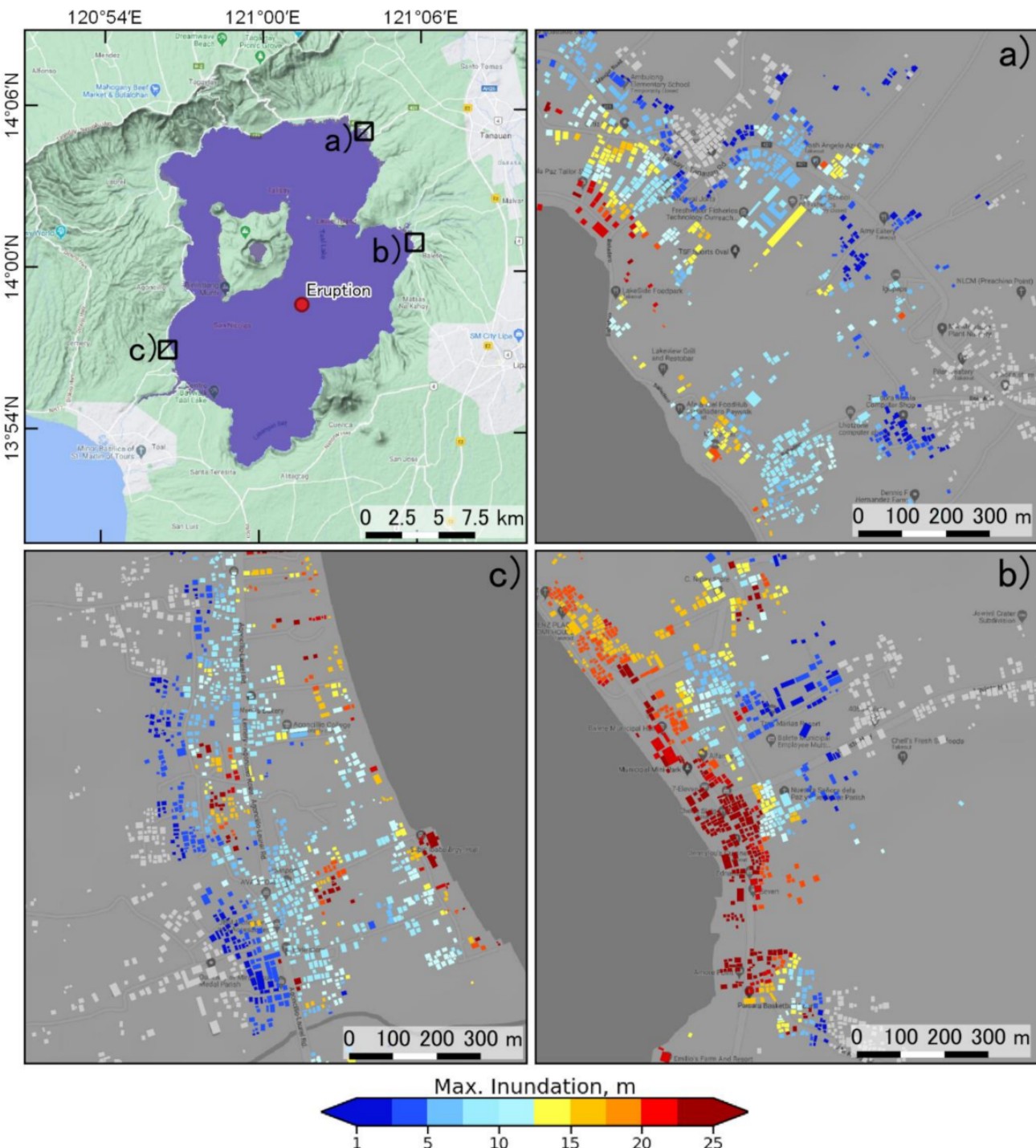

**Figure 10.** Tsunami hazard map showing the maximum inundation depths for the buildings with a probability of 1/100 in some example areas. (**a**) northern coastal area, (**b**) coastal area near to the eruption point, and (**c**) southern coastal area.

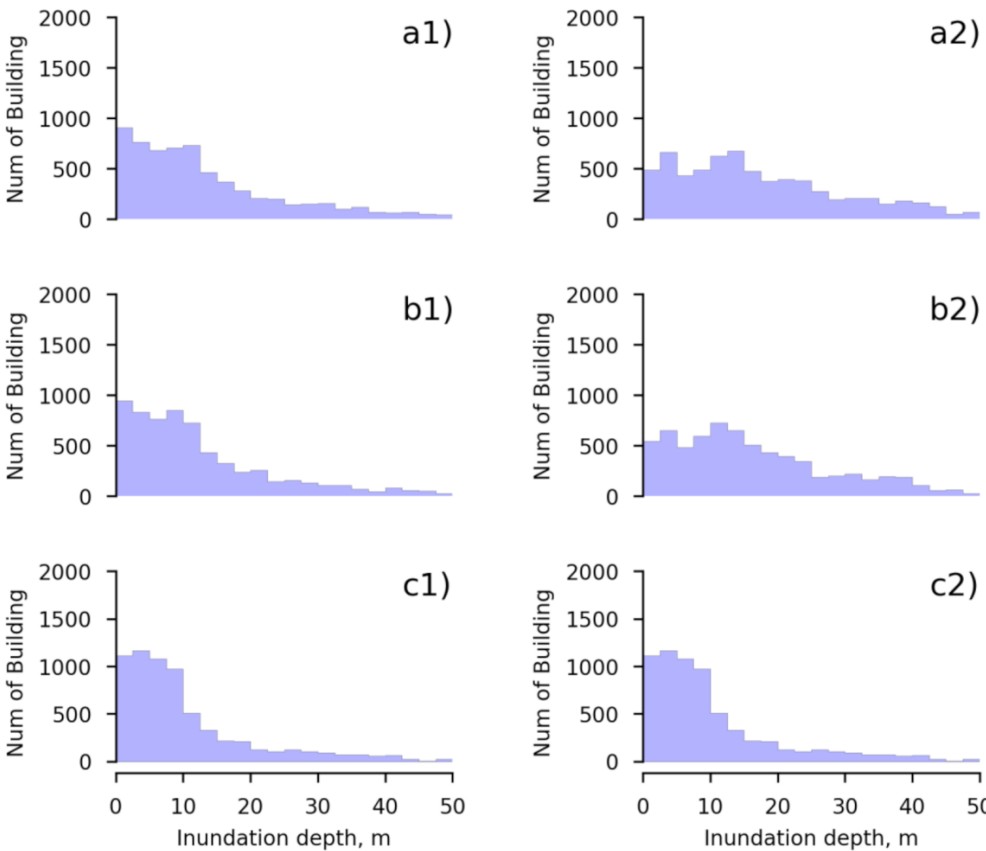

**Figure 11.** Summary of the number of buildings in each inundation depth range of 2.5 m in the whole inundated building around the coastal area of Taal Lake. (**a**) 1/1000 probability, (**b**) 1/500 probability and (**c**) 1/100 probability; the right column corresponds to the northern area and the left column corresponds to the southern area.

These analysis results can be useful for stakeholders on the local side, so that the stakeholders can use the hazard map to be more aware of the hazard level of residential areas. For example, if a house is located in a zone with a probability of approaching one (99%), the people in the house need to evacuate immediately to a safe zone when a tsunami occurs. In contrast, if the probability of a house is approximately zero (0%), the people in the house may need to evacuate only in some limited situations. Policymakers can apply this map for planning residential tsunami safety zones for future tsunami prevention, and they can also use the map for estimating direct disaster losses immediately when a tsunami occurs in the southern basin.

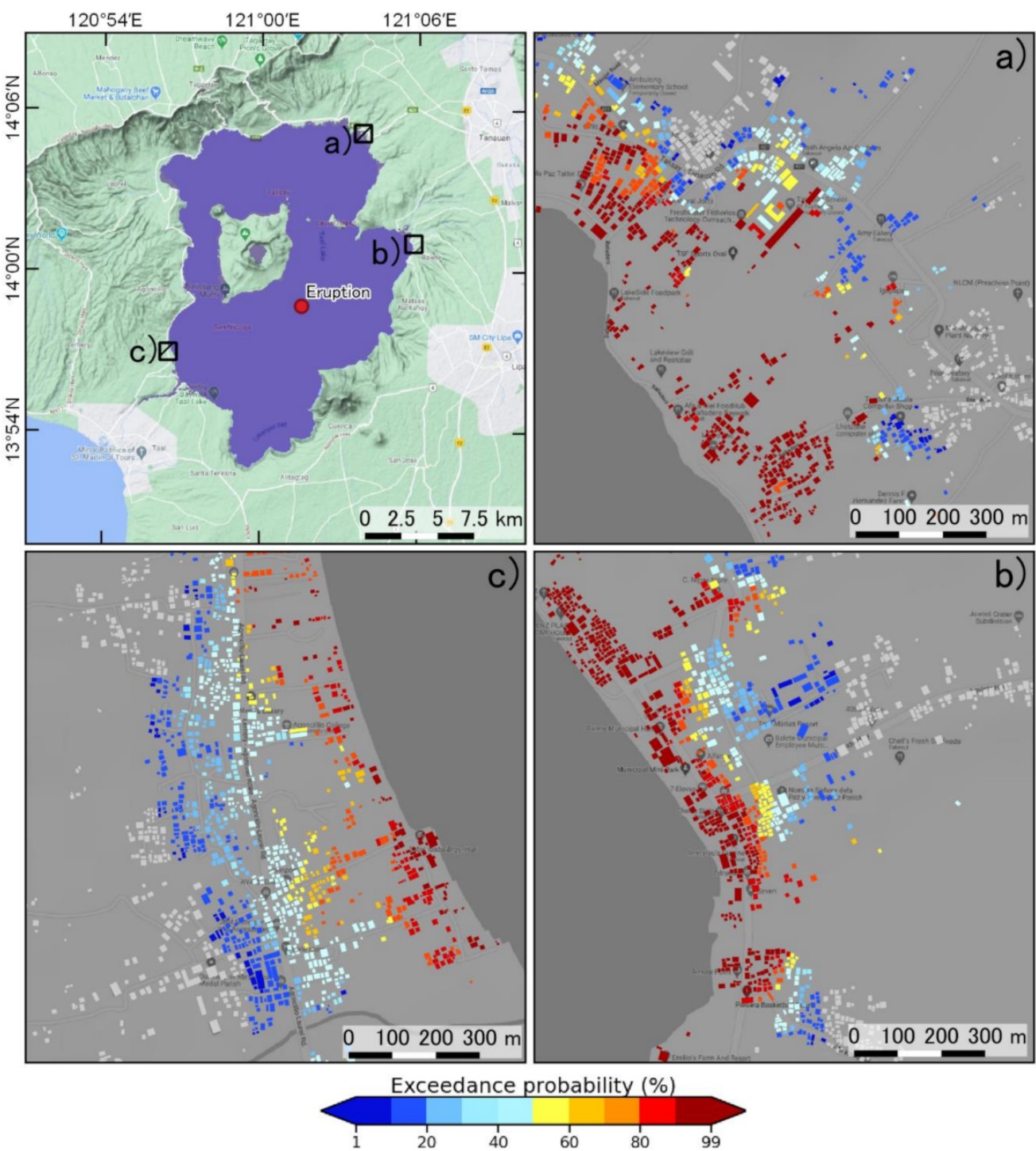

**Figure 12.** Probability of experiencing a tsunami inundation depth of more than 0.5 m in some example areas. (**a**) northern coastal area, (**b**) coastal area near to the eruption point, and (**c**) southern coastal area.

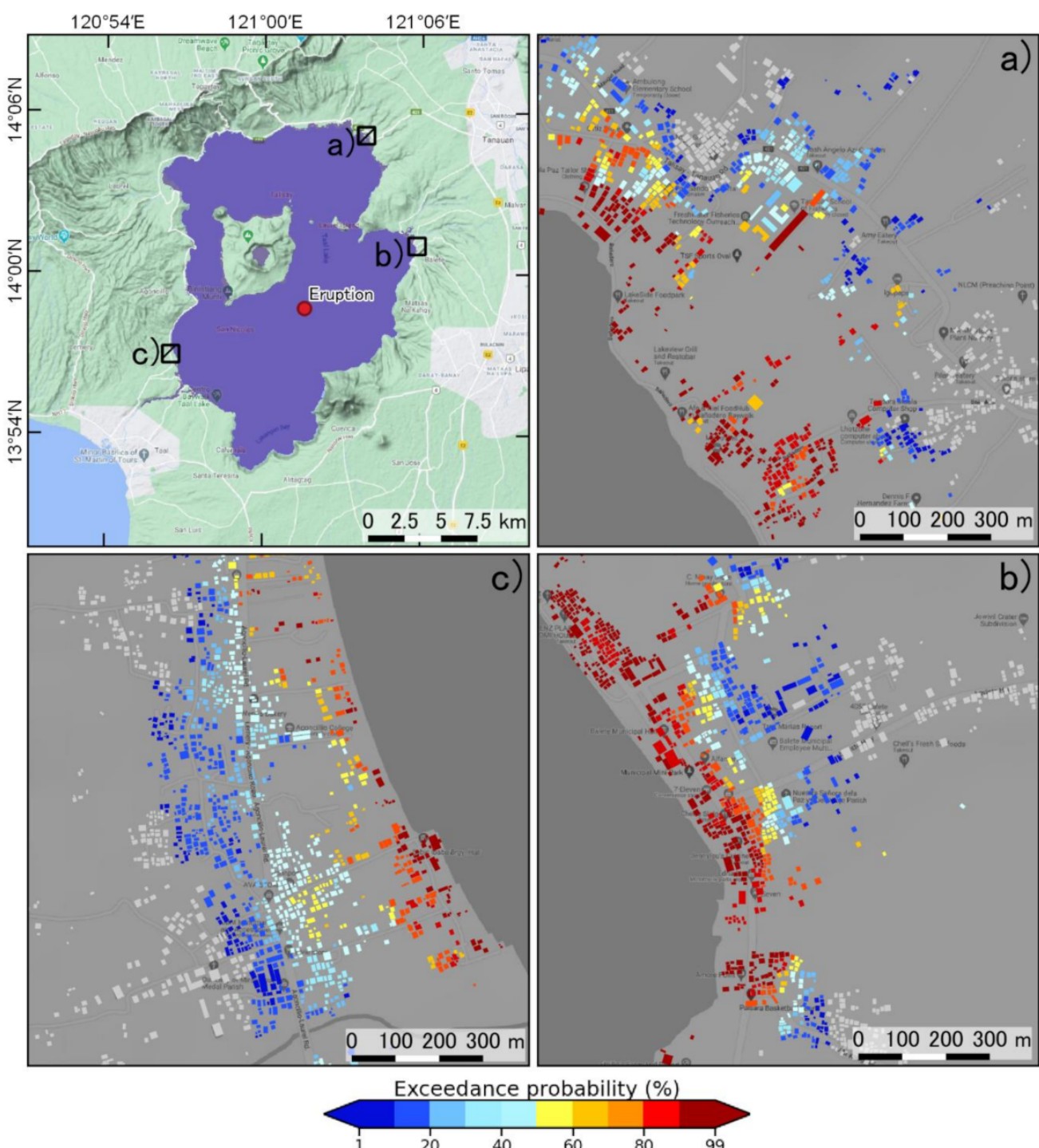

**Figure 13.** Probability of experiencing a tsunami inundation depth of more than 1.5 m in some example areas. (**a**) northern coastal area, (**b**) coastal area near to the eruption point, and (**c**) southern coastal area.

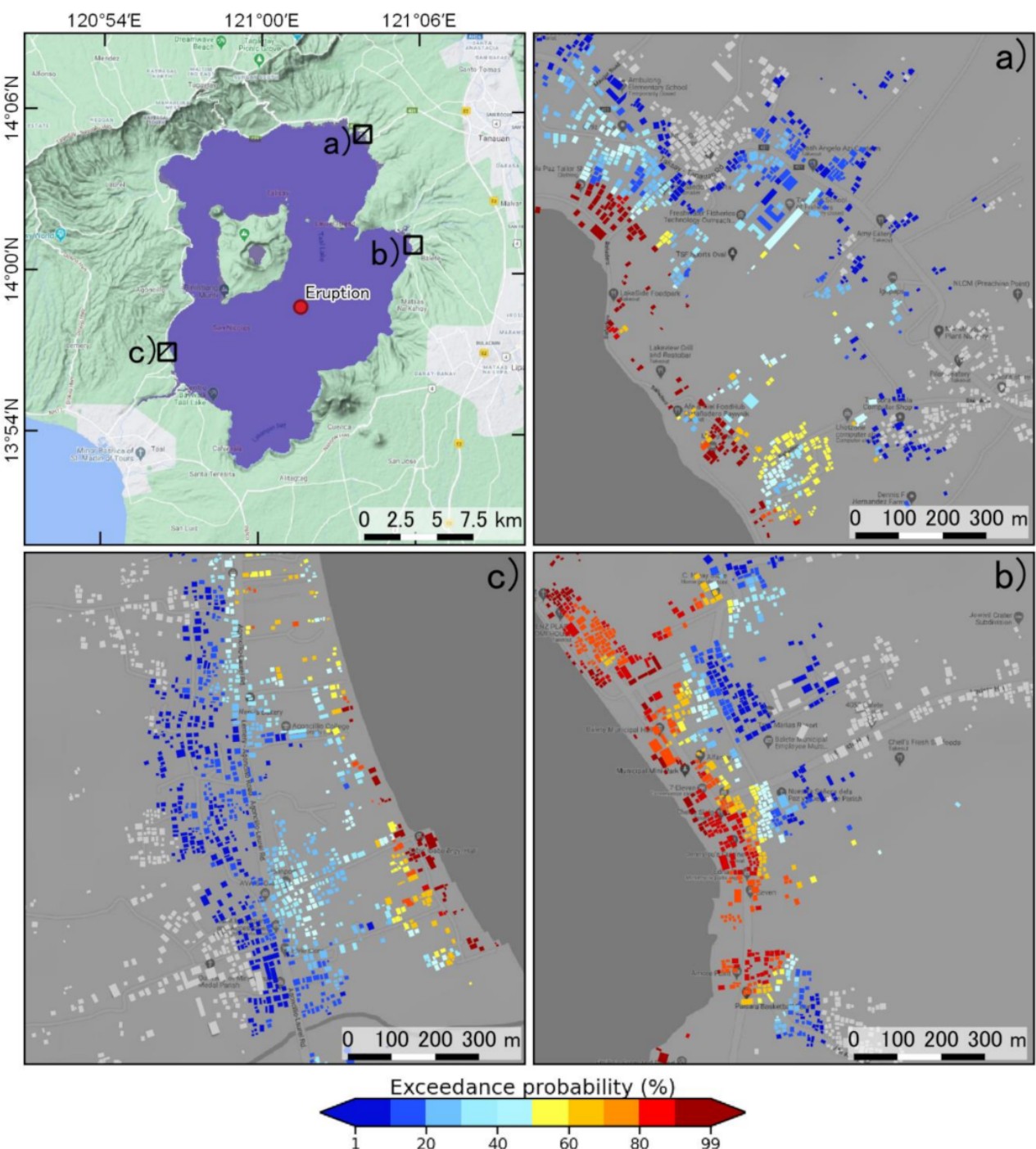

**Figure 14.** Probability of experiencing a tsunami inundation depth of more than 3.0 m in some example areas. (**a**) northern coastal area, (**b**) coastal area near to the eruption point, and (**c**) southern coastal area.

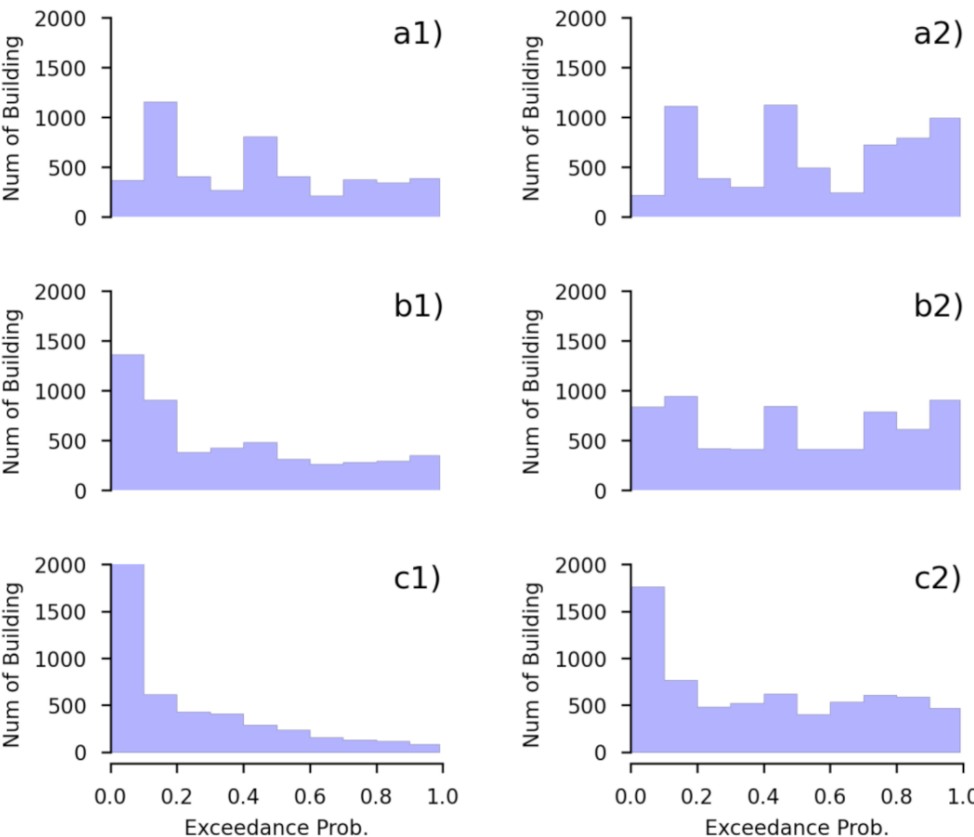

**Figure 15.** Summary of the number of buildings in each exceedance probability range of 0.1 in all the inundated buildings around the coastal area of Taal Lake. (**a**) probability of experiencing a tsunami inundation depth of more than 0.5 m, (**b**) probability of experiencing a tsunami inundation depth of more than 1.5 m and (**c**) probability of experiencing a tsunami inundation depth of more than 3.0 m; the right column corresponds to the northern area and the left column corresponds to the southern area.

## 4. Discussion

Tsunami hazards related to subaqueous volcanic explosions in Taal Lake can occur in the future with an eruption at the bottom of the lake in the underwater caldera. The probabilities of a tsunami hazard with several eruption sizes can be used to develop recommendations on how to address future tsunamis in this area. To consider the largest eruption scenario (the 99th scenario), the highest wave amplitude at the coast of approximately 50 m occurs in the eastern area, while the lowest wave amplitude at the coast of approximately 20 m occurs in the northwest area. This scenario created a runup wave on land that inundated an area of approximately 70 km$^2$, including approximately 13,548 buildings. On the other hand, the smallest explosion scenario (the 1st scenario) presents an inundation area of approximately 0.05 km$^2$, including 50 inundated buildings. The simulation results were limited by the 30-m resolution for the bathymetry and topography. The building was considered a flat surface, which is a limitation of this study. However, we would like to recommend that the building height is input into the model in future studies, and the flow depth would be a great improvement.

Tsunamis observation data from the history of eruptions are limited in terms of providing constraints on the wave height and inundation depth. According to records on the 1716 submarine explosion, the wave inundated the southwestern shore of the lake and reached a height of up to 17 m on land [5]. This submarine explosion event can be assumed to correspond to the 50th scenario in this study. In this study, we provide the probability of building hazards from a subaqueous volcanic explosion generating a tsunami

in Taal Lake with a conditional scenario of hazard occurrence. The tsunami hazard in this study was determined by an explosion empirical formula and tsunami wave propagation model, without calibrations with observed data of the historical event, the tsunami height or inundation depth. However, in the future, after the tsunami height or inundation depth of the historical event are surveyed and recorded, we recommend rerunning the analysis to improve the results.

In this research, tsunami propagation in the lake and on the shore, such as wave breaking and nonhydrostatic pressure terms, was not simulated because of the high computational cost. Another limitation is associated with the model of the explosion itself: a vent with a given size produces a more powerful explosion at shallow depths [20,40]. While the morphology of the crater might be influenced by a combination of vertical and horizontal explosions [41,42], the vent size assumption for explosion magnitude might result in an overestimation in some scenarios. We considered variability only in the vent size, and the only modeled explosion location corresponded to that of the 1716 event in the study area. Studying random variations in the explosion location has been recommended by Paris and Ulvrova [3]. In this paper, we focused only on tsunamis generated by subaqueous volcanic explosions in Taal Lake in the southern basin, while an explosion in the northern basin would generates different patterns of wave heights at the coast, which would be recommended for inclusion in future work. On the other hand, other sources such as pyroclastic density currents flowing into the lake, or landslides triggered by earthquakes, should be integrated into the probabilistic analysis to fill gaps in the tsunami hazard assessment, as explained in Grezio et al. [43].

## 5. Conclusions

Taal volcano is one of the most active and dangerous volcanoes on Luzon Island in the Philippines, and this volcano presents risk mitigation challenges to humans. In its history, eruptions have occurred on the central island, such as the 1749, 1754, 1911 and 1965 eruptions, while the 1716 eruption was a subaqueous volcanic explosion. We simulated different scenarios of subaqueous explosions based on the location of the 1716 event and the tsunami disaster related to this explosion. In this study, a conditional probability tsunami hazard assessment of building exposure generated by subaqueous volcanic explosions is performed for Taal Lake. The scenarios all show that the tsunamis generated by such subaqueous explosions have a large impact on the shores of the lake.

The probability hazard of inundated buildings can be presented by the hazard map. The results of this study, a tsunami hazard probability assessment, present an estimate of the probability of a tsunami hazard impacting urban areas following a subaqueous volcanic explosion in Taal Lake for stakeholders and policymakers. Stakeholders and policymakers can use the results for future tsunami mitigation in the case of a tsunami generated by a subaqueous volcanic explosion.

**Author Contributions:** Conceptualization, K.P., A.S. and F.I.; Data curation, K.P.; Formal analysis, K.P.; Funding acquisition, A.S. and F.I.; Methodology, K.P.; Supervision, A.S.; Validation, K.P.; Visualization, K.P.; Writing—original draft, K.P., A.S. and F.I.; Writing—review & editing, K.P. and A.S. All authors have read and agreed to the published version of the manuscript.

**Funding:** This research was funded by the Willis Research Network (WRN) under the Pan-Asian/ Oceanian tsunami risk modeling project through the International Research Institute of Disaster Science (IRIDeS) at Tohoku University.

**Acknowledgments:** In this study, QGIS software was used to illustrate the spatial data and collect building data from OSM.

**Conflicts of Interest:** The authors declare no conflict of interest.

**Appendix A**

The building data input into the flow model are represented by the Manning coefficient, as shown in Equation (9) in the manuscript. Figure A1 shows that the density of the building was estimated in the model with the same method used for the topography grid. As shown in Figure A1, the green box is the topography grid represented by the DEM data with an area of $dx \times dy$. In the computation grid represented by the topography grid, the three buildings had different locations on the DEM grid. To estimate the building density domain as in the topography grid domain, the sum of the building area in a grid was divided by the area of topography grid ($dx \times dy$). For example, grid C2 was used to estimate the building density with the sum of the area $(B12 + B34)/(dx \times dy) \times 100$, while A2 was zero where there was no building.

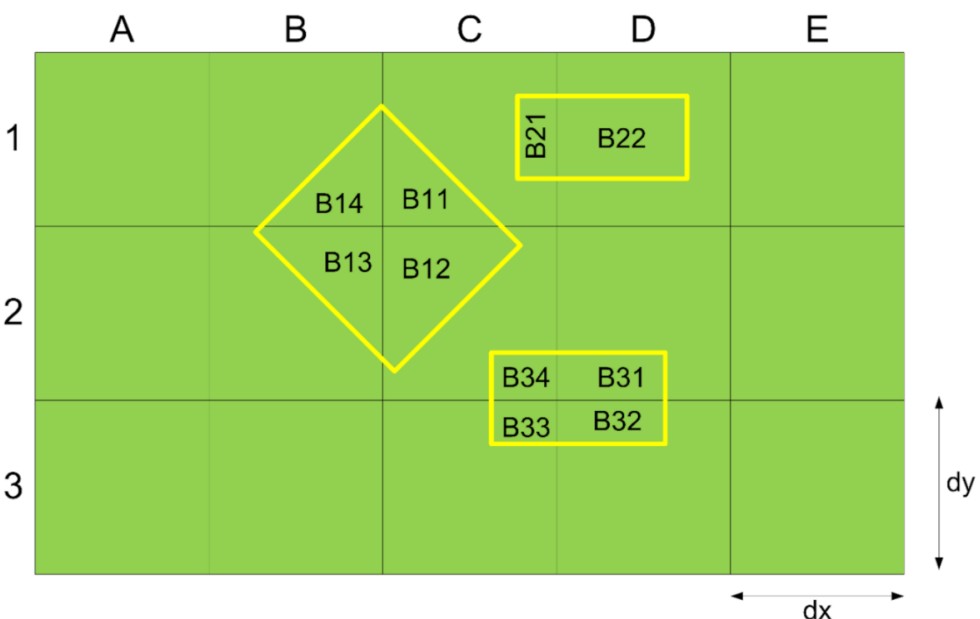

**Figure A1.** Estimating the building density in the topography grid.

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
