# Peer review of "Probabilistic Tsunami Hazard Analysis of Inundated Buildings Following a Subaqueous Volcanic Explosion Based on the 1716 Tsunami Scenario in Taal Lake, Philippines"

_geosciences, doi:10.3390/geosciences11020092_

Round 1

Reviewer 1 Report

Pakoksung et al. (the manuscript) propose a probabilistic tsunami hazard model for building inundation caused by a subaqueous volcanic explosion in Taal Lake, Philippines. They conclude that the results of their study can be used for future tsunami mitigation by stakeholders and policymakers.

The manuscript builds on military research from the 1970’s on underwater explosions to model the initial deformation of the lake surface after a volcanic explosion at the location of the 1716 eruption, for a variety of vent areas. The authors consider 99 scenarios, differing on vent diameter which ranges from 100m to 1000m. Each vent diameter is associated by the authors with a probability, ranging from 1 for the smallest diameter (line 182) to 0.001 for the largest diameter. The resulting tsunami is numerically computed for each scenario of water surface deformation, and the inundation area and number of inundated buildings are computed.

The number of inundated buildings as a function of inundation depth for the different scenarios is fitted by a lognormal distribution, chosen among several candidate distributions. Then, the models for each scenario are combined using the assigned probabilities mentioned above, to derive a tsunami hazard probability map and a tsunami hazard probability curve of inundated buildings (Fig. 10). The authors claim that the tsunami hazard map can be used by stakeholders to assess the hazard level of their residential area, and the map can be used for evacuation management.

Each scenario treated by the authors may be regarded as a deterministic hazard assessment (if an explosion with a vent of this size happens, what is the probability that at a given site the inundation depth will exceed a given value?). This in itself would be a valuable result that could be used for emergency management (immediately after an occurrence, the nearest scenario could be used for a fast estimate of the inundated buildings). However, the meaning of the combination of all the scenarios in the “hazard probability curve” of figure 10 is obscure to me, namely because the meaning of the probabilities used to weight them in the combination is not clear.

The reason to use the lognormal distribution fitted to the building inundation data instead of the modeling results themselves is unclear.  Some statements sound like truisms: “the first scenario (the smallest hazard) has a lower water depth than the 99th scenario (the greatest hazard)” (line 263). How could it be otherwise? On the other hand, some results seem impossible: figure 7 seems to indicate that the 50th scenario inundates more buildings with inundation depths below 1m that the 99th scenario, which has a larger energy. Since the location of the vent is the same, it is not clear how this is possible.  

The most relevant aspect that I found in the paper is the modeling of the initial water surface deformation due to a subaqueous explosion and the forward modeling of the resulting tsunami. Unfortunately that contribution does not add, in my opinion, to the work of Paris and Ulvrova (2019), also for the Taal crater lake, who used the same approach, with the advantage that they modeled for different vent locations.

As a practitioner of probabilistic (seismic) hazard assessment for several decades, I am defeated by the approach taken by the authors, which in several steps I was not able to follow. Hazard communication – the declared goal of the manuscript – requires the adherence to clear and standardized concepts and language. “Probabilistic hazard” must mean the probability that a given intensity of an adverse phenomenon will be exceeded at a specified site during a specified exposure period. For example, the probabilistic tsunami hazard at a site can be a probability of 10% that over a period of 50 years the inundation depth at that site will exceed 5m. None of the results presented in the manuscript resemble such a statement. No mention of an exposure period is made when probabilities are assigned to each scenario (what is the meaning of “the first scenario features the smallest explosion size with a probability value of 1”?), no indication is given of what science is behind those probabilities (even when a Monte Carlo approach is taken, some empirical evidence guides the process). Since the exposed building stock is factored into the calculations, the results in Figure 10 would be better described as "risk" and not "hazard" (if the probability of exceeding a given inundation depth at each site over a given period of time could be estimated soundly).

I have no alternative but to recommend the rejection of the manuscript.

Author Response

Dear Reviewer,

Thank you very much for taking your time to read and recommend my manuscript. please see the attached file.

Reviewer 2 Report

Dear colleagues,

I'm glad to see that the topic of volcanic tsunamis is getting more and more studied. Taal volcano is a good target and you proposed a relevant approach based on combined numerical simulations and probabilistic analysis. It is thus in logic continuity with what I have done on Taal recently (Paris & Ulvrova, 2019) and it could really improve and help tsunami hazard mitigation at Taal volcano. It could even serve as an example for future studies on volcanic tsunami hazard.

However, many corrections are required before it could be published. I have identified 6 major points of concern, some of them being strong limitations:

  1. The resolution of the grid (30 m only) is a serious limitation, especially when you integrate the information on building density as a modified Manning's roughness coefficient (Equation 9).
  2. It is not clear how you define "building inundation". Is the building considered as inundated when it is simply included in a specific range of simulated flow depth? Do you consider also the height of the building compared to the flow depth? In my opinion, the hazard is not the same if the building is just inundated by a water flow less than 50 cm high or if it is fully submerged.
  3. I'm a little bit dubious about by the way you determine the probabilities of each scenario. Please explain the method that is used here in more details.
  4. Scenario 1 corresponds to an explosion energy that is already high (1E15 J) compared to the "low" energy scenario used by Paris & Ulvrova 2019 (3.5E13 J). Scenario 1 is not really a minimum. In my opinion, scenario 1 should be an explosion that generates small waves in the lake but no inundation inland. This corresponds to the threshold of ~3.5E13 joules proposed by Paris & Ulvrova (2019).
  5. I can't see the reason why you want to test 9 different best-fit distributions. It is clear that it has a log-normal trend, and this is what was expected. You can delete this step.
  6. Your conclusions are valid only in the case of an explosion in the southern basin of the lake. Explosions in the northern basin generate different patterns of wave heights at the coast.

For minor corrections, you will find an annotated version of the pdf file. I encourage you to address all these comments and propose a revised version of this original work on a rising topic of interest.

Best regards

Author Response

Thank you very much for taking your time to read and recommend my manuscript. Please see the attached file.

Reviewer 3 Report

This manuscript starts a very useful investigation of inundation probability around the Taal Lake. As the Authors correctly write in the discussion, it is a first step; it is to be considered a preliminary study, more focused on the methodology and on the detection of the best fitting method, rather than on the achievement of an exhaustive and final hazard map (this should be clearly stated also at the beginning). For this reason, I think it deserves to be published after moderate adjustments. I have a few suggestions and some personal curiosity about the approach and models than maybe can be useful to clarify also for other readers not expert, like me, in tsunami modeling.

In general, I think that a hazard is already a probability; so, I suggest avoiding the recurrent use of “probability hazard” or “hazard probability” in the text.

Why do the model create a bore? Is it maybe a crater collapse, rather than an explosion? A gas explosion should produce a spherical (or almost, depending on the crater and conduit shape) expansion and then a rise of the water level. Maybe this aspect should be briefly described.

At the beginning of chapter 2.3, I do not understand if the inundation is only calculated by the wave height with respect to topography. If it is the case, why the Authors introduce the discussion about the roughness and effects of buildings on the water flow?

The probability function of the eruption size from 1 to 0.01 is assumed or is deriving from a statistical analysis of real events (in this case should be cited)? Authors maybe answer to this in the discussions but maybe something should me said in chapter 2.3

In figure 3 the position of the vent in the map seems to not correspond to the profile. In maps, the vent is much closer to the volcano than in the profiles.

In figures 7 and 8, please could you use also different line styles? It is difficult to distinguish only the different colors.

Minor issues

Line 38 – “…principally interesting for evaluating and forecasting”

Line 50 – “The eruption vent was located north of Taal Island and the waves impacted the northern shores of the lake.”

Line 54 – “…eruption occurred in 1922 on Taal Island”

Line 55 - “The 1965 eruption”

Line 91 – It is not necessary to ay again “(Luzon Island, Philippines)”. Already said in the introduction

Line 106 – not clear “involved identifying”

Line 107 – Is 600m the diameter of 1716 vent?

Line 109 – “100 to 1000”

Line 160 – “The grid was obtained”

Line 164 – “and the Open Multi-Processing (OpenMP) platform was applied”

Line 219 – “calculated by Equation 2”

Line 233 – Quite obvious

Line 311 – Usually, the evacuation is organized by policymakers, as the Authors says in Line 315

Line 358 – “... by such a subaqueous…”

Author Response

(The authors gave the same response as above.)

Round 2

Reviewer 1 Report

In my previous review, I highlighted the importance of clarity in hazard communication, specially when a manuscript claims that "these final results can be useful for stakeholders on the local side and policymakers on the government side" (line 285-287). While the authors made a clear effort to adjust the original manuscript to the critical comments I made, I fear that clarity did not increase. I find myself at a loss to understand the meaning of statements such as "if a house is located in a zone with a probability of approaching 0, the people in the house need to evacuate immediately to a safe zone when a tsunami occurs" (lines 287-288). Trying to understand the meaning of the probability referred here, I found the following description: "We produced a tsunami hazard map of building exposure from the probability of the hazard curve in each scenario..." (lines 257-258). These scenarios correspond to different vent sizes a single location (that of the 1716 eruption) and no steps were taken to assign probabilities to each scenario based on empirical data. Ultimately these scenarios are conjectures, not anchored in empirical data (as the authors acknowledge), and I fail to see how a decision by stakeholders or policymakers can be based on conjecture. Even if that were the case, the formulation would be obscure: since the probabilities correspond to different scenarios, upon occurrence of a tsunami which scenario should the stakeholder consider to base their decision? And how is a probability of exceedence near 0 signify danger?  These doubts I was  not able to elucidate reading the manuscript.

I regret to conclude that my main concerns about the manuscript persist after the revision, and I therefore maintain the previous recommendation.

Author Response

Dear Reviewer,

Thank you very much for taking your time to read and recommend my manuscript, in round 2nd. Your recommendations were revised and shown by the yellow highlight. The English language of this manuscript was edited by the American Journal Experts Company.

Best regards,

Author

Reviewer 2 Report

Dear colleagues,

Thanks for addressing all my comments and for your detailed reply. You have indeed revised the manuscript accordingly, but there are still some corrections to do. Editing of English language and style is still required before the contribution can be published (e.g. the introductory sentences are still difficult to understand; same at lines 171-172, 294-295, and 308-312).

Major points :

Point 1 of my former recommendation (grid resolution): thanks for the clarification. This explanation should be developped in the main text. You could even add the fig provided in your reply as an appendix.

Point 2: It would be good to add in the discussion that considering the buildings as "flat" is a limitation, and that taking into account the height of the buildings vs flow height would be a great improvement for future studies.

Point 4: I don't know if we understand each other. I think that your scenario 1 should correspond to the low energy scenario of Paris & Ulvrova (2019), meaning that your scenario 1 should correspond to "no inundation" (no or very few buildings inundated). This point still needs to be clarified and would require that you run additional simulations and change the manuscript accordingly. This is why I still recommend a major revision.

Minor correction:

line 109: "(vent diameter varied from 100 to 1,000 m)"

Author Response

Dear Reviewer,

Thank you very much for taking your time to read and recommend my manuscript, in round 2nd. Your recommendations were revised and shown by the green highlight. The English language of this manuscript was edited by the American Journal Experts Company.

Best regards,

Author

Round 3

Reviewer 2 Report

Dear colleagues,

Thanks for these last efforts. In my opinion the paper is now almost ready for publication. Sentences at lines 310-311 and 330-331 could still be improved.

Best regards